# Conversion of Natural Soil to Paddy Promotes Soil Organic Matter Degradation in Small-Particle Fractions: $\delta^{13}$C and Lipid Biomarker Evidence

**Yuxuan Li [1], Yan Li [1], Yu Zhang [1], Bingzhen Wu [1], Dandan Zhou [1], Hongbo Peng [1,2], Fangfang Li [1,*] and Min Wu [1]**

[1] Yunnan Provincial Key Lab of Soil Carbon Sequestration and Pollution Control, Faculty of Environmental Science & Engineering, Kunming University of Science & Technology, Kunming 650500, China; 15572600335@163.com (Y.L.); li_beerbears@163.com (Y.L.); 18087042170@163.com (Y.Z.); 15388698662@163.com (B.W.); 01yongheng@163.com (D.Z.); mzxb817@163.com (H.P.); minwup@hotmail.com (M.W.)

[2] Faculty of Modern Agricultural Engineering, Kunming University of Science & Technology, Kunming 650500, China

\* Correspondence: fangkust@163.com; Tel./Fax: +86-871-65170906

**Abstract:** The stabilization mechanism of soil organic matter (SOM) has received considerable attention. It is widely accepted that mineral sorption/protection is important for SOM stabilization. However, it remains unclear which organic carbon component is beneficial for mineral protection. We collected soil samples from a paddy field (TP) to compare with natural soil (NS). To illustrate the behavior of different SOM pools and their protection by particles, we separated the soils into different particle-size fractions and then removed the active minerals using an acid mixture (1 M HCl/10% HF). The different carbon pools were analyzed using stable carbon isotopes and lipid biomarkers. Our study showed that acid treatment evidently increased the extractability of free lipids, usually over 60%, which confirmed the predominant role of minerals in SOM protection. For NS, the $\delta^{13}$C values increased with decreasing soil particle sizes and soil depths, indicating that $^{13}$C-enriched SOM was selectively preserved. However, this trend disappeared after cultivation, which was mainly attributed to the combined effects of the input of $^{13}$C-depleted fresh SOM and decomposition of the preserved $^{13}$C-enriched SOM. Meanwhile, based on the degradation parameters of the overall lipid biomarkers, SOM showed higher degradation states in clay and silt fractions than in the sand fraction before cultivation. It is possible that the small particle-size fractions could selectively absorb highly degraded SOM. The clay-associated SOM showed a low degradation state, but its carbon content was low after cultivation. We propose that the previously protected SOM was degraded after cultivation and was replaced by relatively fresh SOM, which should be carefully monitored during SOM management.

**Keywords:** free lipids; $^{13}$C natural abundance; mineral removal; long-term cultivation; organic matter decomposition





## 1. Introduction

Soil organic matter (SOM) is the largest carbon reservoir in terrestrial ecosystems and plays a pivotal role in mitigating global climate change [1]. The main mechanisms that researchers frequently refer to regarding SOM stabilization include (i) resistance against decomposition due to the inherent chemical characteristics of biomolecules [2,3]; (ii) less accessibility due to the development of soil aggregates separating organic matter (OM) from decomposers [4,5]; and (iii) stabilization of OM in organo-mineral associations via physicochemical interactions [6,7]. Most studies found that aggregates and organo-mineral complexes were key components for controlling the long-term stabilization of SOM [5,8]. For example, degradable "old" SOM was well preserved in clay and silt

fractions [9,10]. However, some studies found that SOM stabilization was dependent on its chemical composition [11]. Therefore, researchers have not reached an agreement on which parameter plays a dominant role, and thus practices in soil carbon sequestration are greatly impeded.

Intensive agricultural activities can accelerate SOM degradation, resulting in soil quality declines and $CO_2$ discharge into the atmosphere [12]. The input of fresh residues can increase the carbon content; however, it also may accelerate carbon turnover, particularly in a nutrition-limited system. This is why even with substantial carbon input, such as organic fertilizers, a general carbon deficit is reported in agricultural soils [13,14]. Considering SOM stabilization by mineral particles, it is still unknown how the mineral-associated SOM responds to tillage activities. For example, some studies suggested that clay-associated carbon showed lower carbon mineralization after cultivation compared to carbon associated with large-size fractions [15,16]. However, Churchman et al. [17] found that more carbon was lost from clay-size fractions compared to sand-size fraction due to breakup of aggregates. Keiluweit et al. [18] also found that mineral-associated SOM would be released and degraded with inputs of more labile carbon.

Considering the diverse origins and quick turnover of SOM, especially in intensive agricultural systems, the analysis of bulk SOM may not always be persuasive. Molecular biomarkers and stable carbon isotope techniques are useful tools for investigating the source and turnover of SOM [19]. Lipids are widely used biomarkers to distinguish plant-derived and microbial-derived sources [20] and can remain well preserved in soil for decades to millennia [20]. It is generally accepted that short-chain lipids (<C20) (including short-chain and iso-alkanoic acids, alkanes and alkanols) are derived from microorganisms [21], while long-chain lipids (including long-chain alkanoic acids, alkanes, alkanols and phytosterol) are derived from higher plants [22]. Therefore, lipid molecular biomarkers and carbon isotopic analysis are complementary and can be applied to better understand the fate and degradation of SOM.

Thus, we jointly applied lipid molecular biomarkers and carbon isotopic analysis to investigate the turnover of mineral-protected SOM. The acid mixture of 10% hydrofluoric acid (HF) and 1 M hydrochloric acid (HCl) was used to identify the extent of mineral protection [23,24]. Soil samples were separated into different particle-size fractions, with an emphasis on SOM composition as affected by agricultural activities. The sampling area was a paddy field with over 50 years of cultivation. The original vegetation was a broad-leaved forest that was dominated by *Aporosa yunnanensis*. The findings of this study will provide essential information for understanding and managing the turnover of SOM in agricultural soils.

## 2. Material and Methods

### 2.1. Soil Samples Collection and Pretreatment

The sample site was located near Dadugang Village in Xishuangbanna, Yunnan province, China. The climate is tropical and humid. We collected two soil samples from the same region in March 2015. The natural soil (NS) without tillage was abundant in the broad-leaved forest (101°1′1″ E, 22°14′36″ N). The paddy soil (TP) (101°0′56″ E, 22°14′40″ N) was transformed from NS and has been cultivated for over 50 years according to a survey of local farmers (long-term single-crop cultivation of rice). The soil is a Haplic Ferralsol (according to http://www.soilinfo.cn/map/index.aspx, accessed on 2 April 2024); this soil is sand clay and loamy in texture (detailed information is described in Figure S4). Both soils were sampled with different depths: TS (topsoil, 0–20 cm), SS (subsoil, 20–40 cm) and DS (deep soil, 40–60 cm). Five-point sampling methods were used three times for soil sampling at each site as three parallel samples for subsequent experiment. The procedures of detailed sampling were the same as previously reported [24]. The visible plant residues in the soil samples were removed manually, and soil samples were air-dried and passed through a 2 mm sieve. The pH values in all soils were lower than 6.

### 2.2. Soil Particle-Size Fractionation

The dried and sieved soils were mainly separated into three particle-size fractions: clay (<2 μm), silt (2–53 μm), and fine sand (53–250 μm) [25]. The mass percentage of >250 μm fraction was less than 7% and this fraction almost contained plant residue without minerals. Considering that this study focused on the effect of minerals on different SOM pools in long-term cultivation, this fraction was not considered. The soils were mixed with deionized water at a ratio of 1:5 (*w:v*) and ultrasonically dispersed (output energy if 22 J mL$^{-1}$). According to our previous study [26], the mixture was first passed through a 250 μm mesh sieve. The filtrate was further dispersed with an ultrasonic power of 47.5 W for 10 min. The suspension was then passed through a 53 μm mesh sieve. The particles on the sieve were collected and noted as the fine sand fraction (53–250 μm). The filtrate was centrifuged at $700 \times g$ for 4 min. The settled particles were labeled as the silt-size fraction. To obtain the clay fraction, the supernatant was repeatedly centrifuged at $4000 \times g$ until the supernatant was clear. All fractions were freeze-dried and ground through a 100-mesh sieve for further analysis.

The extent of mineral protection is indicated by the differences or the percentage in amounts of lipids between before and after acid treatment. To investigate the extent of mineral protection, aliquots of the above soil particle-size fractions were pretreated with acid mixtures of 1 M HCl/10% HF to remove reactive minerals [24]. The soil samples were mixed with the acids at a rate of 1:2 (*w:v*) and shaken for 2 h. The mixture was then centrifuged for 15 min. This process was repeated seven times, and then the particles were washed with deionized water several times. The remaining residue was freeze-dried, weighed and ground through a 100-mesh sieve for further analysis. The supernatant was also collected, freeze-dried and weighed.

### 2.3. Extraction and Detection of Lipid Biomarkers in Soils

All the above samples were sequentially extracted using organic solvents for biomarker extraction [24]. Soil particles (1~20 g) were sonicated and extracted sequentially with 30 mL of dichloromethane, dichloromethane/methanol (1:1, *v:v*), and methanol, each for 15 min. The extracts were filtered, mixed, and concentrated until dry under $N_2$.

Before the detection, the dried extracts were re-dissolved in 1~5 mL of dichloromethane/methanol (1:1, *v/v*). Dissolved extracts (100 μL) were dried with $N_2$ and then derivatized with a mixture of *N*,*O*-*bis*-(trimethylsilyl) trifluoroacetamide (BSTFA) and pyridine (9:1, *v:v*) for 3 h at 70 °C. After cooling, all the derivatized solutions were analyzed using gas chromatography–mass spectrometry (GC-MS) (Agilent Technologies, Inc., Palo Alto, CA, USA, 7890A GC equipped with a 5975C quadrupole mass selective detector). Trimethylsilyl derivatives of n-heptadecanoic acid and ergosterol were used as external standards. The main temperature procedure was as follows: the original temperature was 65 °C for 2 min, and it was increased to 300 °C at a rate of 6 °C min$^{-1}$ and then maintained at 300 °C for 20 min. The MS was operated at an ionization energy of 70 eV and scanned from 50 to 650 Da. It is generally accepted that short-chain lipids (<C20) are derived from microorganism [21], while long-chain lipids are derived from higher plants [22]. Therefore, *n*-alkanoic acids/alkanols/alkanes in the range of C14 to C19 and iso-alkanoic acids (C16:1, C17:1 and C18:1) were used to represent microbial-derived lipids and *n*-alkanoic acids/alkanols/alkanes in the range of C20 to C32 and phytosterol (campesterol, stigmasterol and beta.-sitosterol) were used to represent plant-derived lipids.

### 2.4. Stable Carbon Isotopic Analyses in All Soil Particle-Size Fractions

The stable carbon isotope ratio (δ$^{13}$C) of the bulk samples before acid treatment was determined using an isotope ratio mass spectrometer (Delta V Advantage, Thermo Fisher Scientific, Waltham, MA, USA) connected to an elemental analyzer (FLASH 2000HT, Thermo-Fisher Scientific with a precision of <0.05‰) (EA-IRMS). Aliquots of soil samples (2–20 mg, calculated based on 40 μg C per capsule) were weighed into capsules. The δ$^{13}$C

value of each sample was calibrated against NIST glutamic acid (RM 8573), which was reported relative to Vienna Pee Dee Belemnite (VPDB).

*2.5. Calculation and Date Analysis*

The free lipid content after acid treatment is calculated as follows:

$$C_{after} = C_{soil} \times (W_{after} \div W_{before}) + C_{supernatant} \times (W_{supernatant} \div W_{before})$$

where $C_{after}$ is the free lipid content after acid treatment, $C_{soil}$ is the free lipid content measured in the soil after acid treatment, $C_{supernatant}$ is the free lipid content in the freeze-dried supernatant, $W_{before}$ is the weight of the soil before acid treatment, $W_{after}$ is the weight of the soil after acid treatment, and $W_{supernatant}$ is the weight of the supernatant after freeze-drying.

The averages and standard deviations were calculated from three parallel samples for each treatment. The differences in lipids before and after cultivation were compared by an independent sample *t*-test with IBM SPSS statistics version 22.0 (IBM Corporation, Armonk, NY, USA). The differences in degradation parameters or $\delta^{13}C$ values among all soil samples were analyzed by one-way ANOVA.

## 3. Results and Discussion

### 3.1. The Protection of Lipids by Mineral Particles

As presented in Figure 1 and Figure S1, the lipid biomarker content detected in clay was generally higher than the other fractions because of their larger surface area compared to the other fractions. Acid treatment can destroy organo-mineral complexes or aggregates to enhance the extractability of lipids [23]. The comparison of the lipid biomarker content before and after acid treatment suggested the extent of mineral protection. As reported in previous study [24], the extractable lipids mainly included alkanoic acids, alkanes, alkanols, and steroids, and alkanoic acids were more abundant than other lipid compositions. The percentage of alkanoic acids protected by minerals accounted for up to 98% of the overall *n*-alkanoic acids (Figures 1A and S2A), probably because of the strong interaction between *n*-alkanoic acids and reactive minerals via ligand exchange in acidic soils (pH 4.5–5.4 in our study). In addition, the protection of *n*-alkanoic acids by minerals in most soils was generally better in clay and silt fractions than in the sand fraction (Figure S2A), whereas this phenomenon was not observed for other lipid compositions. Compared to other lipid compositions, *n*-alkanoic acids may be more likely to be protected through adsorption. In general, the clay fraction included more reactive minerals than the other fractions. Thus, the extents of *n*-alkanoic acids protected by minerals in the clay fraction were relatively higher than those of the other combinations (Figure S2).

Compared to alkanols and steroids, the protection of *n*-alkanes by minerals was generally better, mostly over 60%. The protection percentages of alkanes were higher in fine sand fractions than those of clay (Figure 1B), which is opposite to that of *n*-alkanoic acids. The strong hydrophobicity of *n*-alkanes may have facilitated their physical occlusion in soil aggregates. This process is not expected for the weak hydrophobic chemicals of alkanols and steroids, and thus their protection by mineral particles was much lower, generally below 30%.

### 3.2. Fresh SOM Input after Cultivation, as Suggested by Bulk $\delta^{13}C$ Values

3.2.1. $\delta^{13}C$ Enrichment in Small Particles

Carbon isotope analysis is a powerful technique for investigating SOM turnover. Before cultivation, the $\delta^{13}C$ values of SOM increased (less negative) with a decreasing particle size (Figure 2). This result indicated that more $^{13}C$ SOM was enriched in smaller-size fractions. Three possible processes may be relevant to this observation. Firstly, previous studies have suggested that microbes may preferentially degrade $^{12}C$ fractions during SOM decomposition, resulting in $^{12}C$ release and $^{13}C$ enrichment in the remaining SOM [19,27].

However, the bioavailability of SOM for small particles is lower than that for larger particles, and thus, more significant $^{13}$C enrichment in small particles could not be explained solely by selective degradation of $^{12}$C fractions.

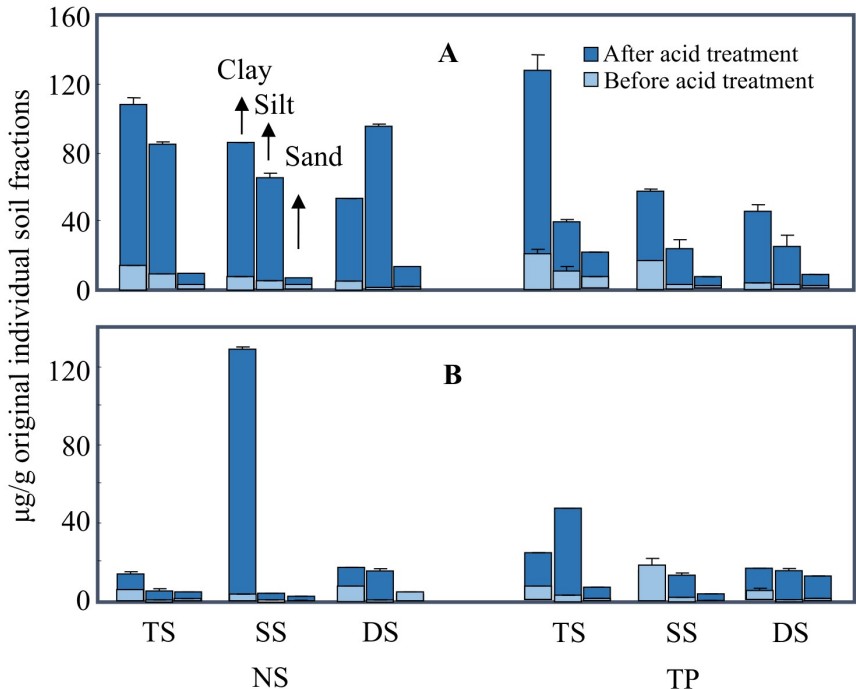

**Figure 1.** The concentration of alkanoic acids (**A**) and alkanes (**B**) before and after acid treatment. NS-natural soil, TP-tillage paddy. Soil layers are categorized as TS (0–20 cm), SS (20–40 cm), and DS (40–60 cm). Considering the mass loss after mineral removal, the calculated concentrations in this study were based on the original mass of soil before mineral removal (mg g$^{-1}$ original soil). Error bar means standard deviation (*n* = 3).

Secondly, it is widely accepted that microbes utilize energy-rich compounds (e.g., carbohydrates and proteins) that are generally enriched in $^{13}$C during their anabolism [28]. Therefore, the microbial residues were enriched in $^{13}$C, and their δ$^{13}$C values were generally higher than those in plant-derived SOM [28,29]. Previous studies have also suggested that microbes are more abundant on small particles than on large particles [30]. According to our measurement of lipid biomarkers, the microbial-derived lipid content increased with decreasing particle size (Figure 3). In addition, the microbial-derived lipid content contributed to over 60% of SOM. Thus, we believe that microbial-derived SOM was enriched on small particles, which is one of the reasons for the $^{13}$C enrichment in small particles. Notably, compared with NS, the δ$^{13}$C values of SS and DS in TP did not show a significant increase with decreasing particle size. This might be explained by the lower accumulation of microbial-derived OM during the cultivation (compared to plant-derived OM, Figure 3).

Thirdly, the selective adsorption of degraded SOM may also result in $^{13}$C enrichment. The selective sorption and fractionation of SOM on particles have been investigated intensively in the literature [31,32]. Evidence has been provided that mineral particles may selectively adsorb organic moieties enriched with $^{13}$C [33,34]. Considering the larger surface area of small particles, their higher sorption may have resulted in more significant $^{13}$C enrichment. The adsorbed molecules may include both microbial-derived and plant-derived molecules.

The latter two possibilities will be further analyzed in the following discussion.

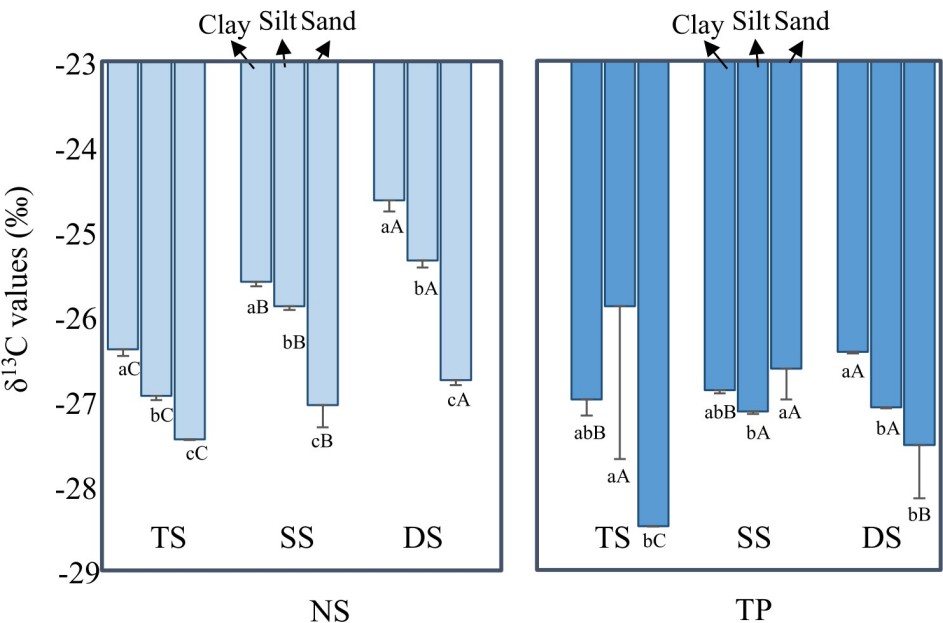

**Figure 2.** The variations in carbon isotope ratio ($\delta^{13}$C) values in clay, silt and sand fractions. Soil layers are categorized as TS (0–20 cm), SS (20–40 cm), and DS (40–60 cm). NS-natural soil, TP-tillage paddy. Different lowercase letters indicate significant difference among different particle-size fractions in same soil depth at $p < 0.05$. Different capital letters indicate significant difference among different soil depths in same particle-size fraction at $p < 0.05$.

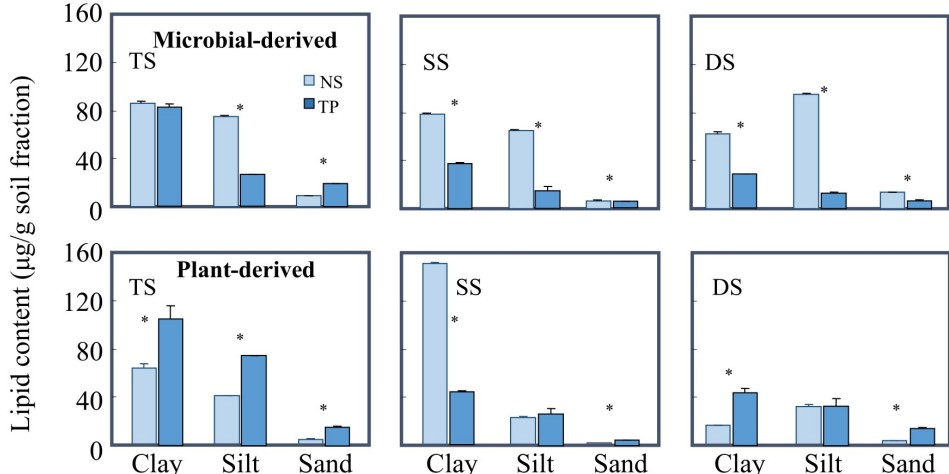

**Figure 3.** The change in microbial-derived lipids and plant-derived lipids of different soil-size fractions. NS-natural soil, TP-tillage paddy. Soil layers are categorized as TS (0–20 cm), SS (20–40 cm), and DS (40–60 cm). * represents significant difference at $p < 0.01$. Microbial-derived lipids mainly include n-alkanoic acids/alkanols/alkanes in the range of C14 to C19 and iso-alkanoic acids (C16:1, C17:1 and C18:1). Plant-derived lipids mainly include n-alkanoic acids/alkanols/alkanes in the range of C20 to C32 and phytosterol (campesterol, stigmasterol and beta.-sitosterol).

### 3.2.2. $\delta^{13}$C Enrichment with Soil Depth

In natural soils, for the same particle-size fraction, the $\delta^{13}$C values of SOM increased with soil depth. Some studies have reported that the increase in $\delta^{13}$C values in deep soil is partly attributed to the progressive degradation of SOM with soil depth [19,35]. However, the C/N ratios, associated with microbial activities, did not change with soil depth before cultivation (Table S1). Thus, SOM degradation may not contribute to enrichment in the $^{13}$C in deep soil.

The downward transport of the degraded SOM, which was enriched with [13]C fractions, may also result in [13]C enrichment with soil depth. However, considering the selective adsorption of the degraded SOM by clay particles, as discussed earlier, the vertical transport of degraded SOM may be significant only when their sorption to clay reaches saturation. This is highly unlikely for soils with a limited OC content in the investigated area.

Therefore, we suggest that the significant input of fresh SOM may be a primary contributor to the depleted [13]C values in the topsoil. Earlier studies have reported that the combustion of [13]C-depleted fossil fuels since industrialization has decreased the $\delta^{13}C$ values of atmospheric $CO_2$, resulting in lower $\delta^{13}C$ values of SOM in surface soil than in deeper soils [28]. However, [13]C enrichment with soil depth became less obvious after cultivation, and the $\delta^{13}C$ values were lower than the corresponding natural soil fractions (Figure 2). As discussed earlier, fresh SOM has a relatively low $\delta^{13}C$ composition [36]. Intensive agricultural activities may have resulted in an increased SOM input in cultivated soils. As discussed in the next section, plant-derived lipids were dominant in tillaged soils (Figure S3). Therefore, protecting the new organic matter could be a key strategy for SOM management in agricultural systems.

### 3.3. The Degradation of Mineral-Protected Lipids

The SOM degradation parameters (e.g., the carbon preference index of long-chain alkanoic acids, $CPI_{AF\geq20}$) were only calculated based on lipid biomarkers after acid treatment, because reactive minerals may decrease the extractability of lipid biomarkers. The missed $CPI_{AF\geq20}$ values in Figure 4 were due to the undetected odd-numbered carbons of the *n*-alkanoic acids, possibly because of the dominance of lowly degraded SOM. Larger $CPI_{AF\geq20}$ values suggest a lower degree of degradation [8,37]. For example, the $CPI_{AF\geq20}$ values of the sand fraction in surface soil were higher than those of silt and clay fractions (Figure 4), indicating a lower degradation extent of SOM in sand.

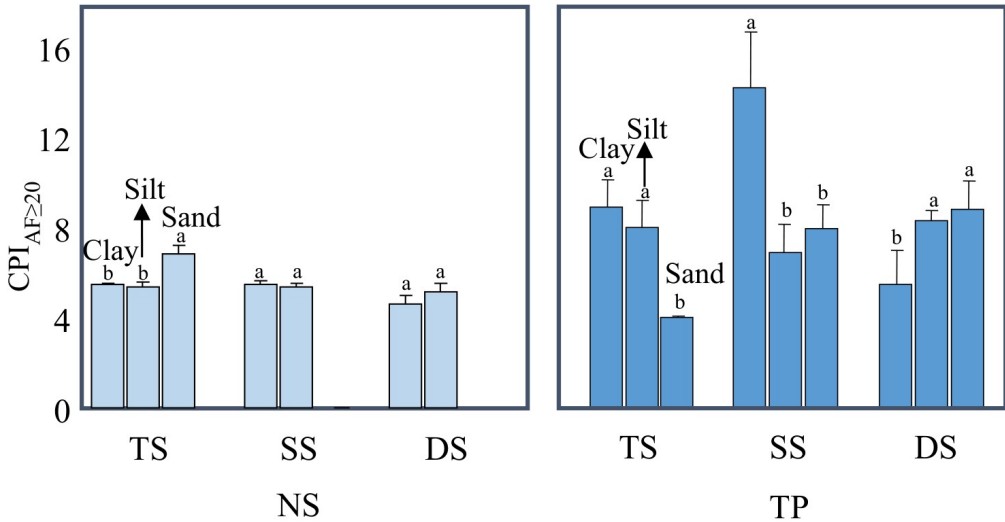

**Figure 4.** The degradation index of lipid composition in clay, silt and sand fractions. $CPI_{AF\geq20}$ indicates the characteristic even-over-odd chain-length predominance of n-alkanoic acid with over $C_{20}$ carbons, $CPI_{AF\geq20} = ((C_{20}\sim C_{30}) + (C_{22}\sim C_{32}))_{even}/2 \times (C_{21}\sim C_{31})_{odd}$; NS-natural soil, TP-tillage paddy. Soil layers are categorized as TS (0–20 cm), SS (20–40 cm), and DS (40–60 cm). Different letters indicate significant difference among different particle-size fractions at the same soil depth at $p < 0.05$.

We proposed in Section 3.2.1 that [13]C-enriched SOM in small particles may be from both microbial- and plant-derived SOM. According to our analysis of lipid biomarkers, it seems unlikely that microbial-derived SOM contributes to this [13]C enrichment. For example, as suggested by microbial-derived lipids (Figure 3) and C/N (Table S1), microbial-derived lipid contents did not change with soil depth. We speculated that [13]C enrichment was mainly due to selective adsorption of [13]C compositions by small particles instead of

SOM degradation or microbial-derived SOM. Similarly, the relatively higher content of smaller particles (Figure S4) in lower soil layers may be responsible for the $^{13}$C enrichment in deeper soils.

After cultivation, the CPI$_{AF \geq 20}$ values of sand fractions in all TP soils depths showed a higher degree of degradation compared to corresponding fractions in NS (Figure 4), while the carbon content of TP sand fractions showed an increasing trend compared to that of NS (Figure 5). Generally, the coarser fractions contained inputs of fresher OM [38]. Although a continuous input of fresh biomass is expected during cultivation, intensive agricultural activities (plowing and fertilizing) may have promoted the degradation of SOM after exposure to oxygen, especially in the presence of reactive Fe [6,39]. Moreover, the CPI$_{AF \geq 20}$ values for clay and silt fractions at all TP soil depths evidently increased compared to corresponding fractions in NS (Figure 4). This showed that the mineral-associated free lipid degradation decreased after cultivation, which implies the input of fresh OM to the mineral-associated OM. Interestingly, the carbon contents in clay and silt fractions were reduced at all TP soil depths compared to corresponding fractions in NS, indicating that fresh OM input did not increase mineral-associated organic carbon contents. On the one hand, the bonds in the organic–inorganic complex would be broken by small molecule organic acids in the fresh OM, resulting in the loss of organic carbon [40]. On the other hand, small-molecule sugars in fresh OM would stimulate soil microbial activity, causing more losses of mineral-associated carbon (priming effect) [40]. We thus propose that the previously protected SOM was degraded after cultivation and was replaced by relatively fresh SOM. In addition, we observed that the loss of carbon content in the silt fractions was higher than that in the clay fractions (Figure 5), indicating that SOM was better stabilized in clay than in silt. Clay may adsorb more fresh SOM compared to silt because of its larger surface area, as confirmed by its higher CPI$_{AF \geq 20}$ values (TS and SS in TP, Figure 4).

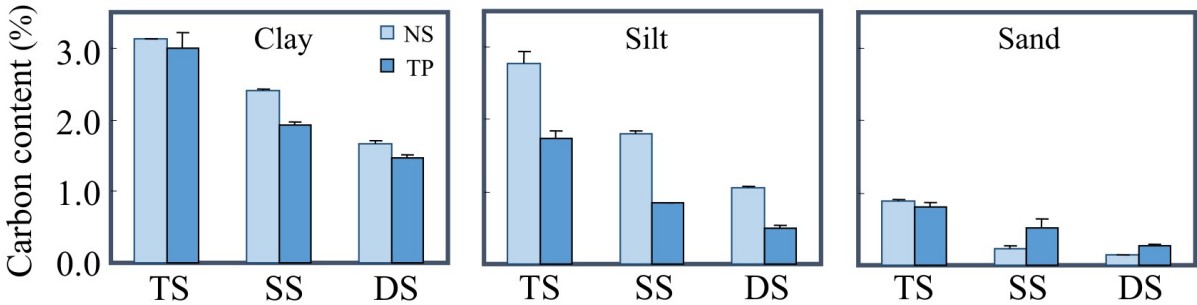

**Figure 5.** The content of soil organic carbon in clay, silt and sand fractions before and after cultivation. NS-natural soil, TP-tillage paddy. Soil layers are categorized as TS (0–20 cm), SS (20–40 cm), and DS (40–60 cm).

## 4. Conclusions

The composition and degradation of SOM associated with soil particle size and soil depth were altered by cultivation to various extents. The $\delta^{13}$C values increased with decreasing particle sizes and soil depths before cultivation, suggesting the accumulation of $^{13}$C-rich compositions during the decomposition process. Cultivation generally led to a decrease in microbial-derived lipid contents and an increase in plant-derived lipids, indicating that microbial-derived OM was more susceptible to long-term cultivation. Based on lipid-related information (e.g., CPI$_{AF \geq 20}$), the clay-associated SOM showed a lower degradation state, whereas its carbon content decreased after cultivation. This study emphasizes that agricultural activities result in abundant input of fresh SOM. The sorption of both microbial-derived and plant-derived SOM on the particles is essential for their stabilization. However, it is also possible that the degradation of SOM previously adsorbed on small particles may be accelerated during agricultural activities, which should be carefully monitored for the management of SOM.

**Supplementary Materials:** The following supporting information can be downloaded at: https://www.mdpi.com/article/10.3390/agronomy14040818/s1, Table S1. The N, H, S content and C/N ratio in SOM before acid treatment. Figure S1. The concentration of alkanols and steroids before and after acid treatment. Figure S2. The percentage of alkanoic acids (A) and alkanes (B) associated with minerals in overall alkanoic acids and alkanes detected in different soil fractions. Figure S3. The percentage of microbial-derived lipid and plant-derived lipid of different soil size fractions in natural soil (A) and tillage paddy (B). Figure S4. The percentage of clay, silt and sand fraction in soils. NS-natural soil, TP-tillage paddy.

**Author Contributions:** Conceptualization, F.L.; Methodology, Y.L. (Yuxuan Li), Y.L. (Yan Li), Y.Z., B.W. and F.L.; Investigation, Y.L. (Yuxuan Li); Resources, F.L.; Data curation, Y.L. (Yuxuan Li), Y.L. (Yan Li) and F.L.; Writing—original draft, Y.L. (Yuxuan Li); Writing—review & editing, D.Z., H.P., F.L. and M.W.; Supervision, D.Z., H.P., F.L. and M.W.; Project administration, F.L.; Funding acquisition, F.L. All authors have read and agreed to the published version of the manuscript.

**Funding:** This research was supported by the National Natural Science Foundation of China (42267028, 42130711 and 42167030), and Yunnan Major Scientific and Technological Projects (202202AG050019).

**Data Availability Statement:** Data is contained within the article and Supplementary Material.

**Conflicts of Interest:** The authors declare no conflict of interest.

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
