# Peer review of "Conversion of Natural Soil to Paddy Promotes Soil Organic Matter Degradation in Small-Particle Fractions: δ13C and Lipid Biomarker Evidence"

_agronomy, doi:10.3390/agronomy14040818_

Round 1

Reviewer 1 Report

Comments and Suggestions for Authors

Review on manuscript “Long-Term Cultivation Results in Soil Organic Matter Degradation in Small-Particle Fractions»

Authors: Yuxuan Li , Yan Li , Yu Zhang , Bingzhen Wu , Dandan Zhou , Hongbo Peng, Fangfang Li,  and Min Wu

The authors studied changes in the content and composition of SOM in different granulometric fractions of soil as a result of long-term cultivation. The problem of preservation of organic matter in soil under intensive cultivation is very important in the modern world, as soil is one of the most important components of the biosphere, participating in the global carbon cycle. Therefore, the work done by the authors will be of interest to readers.

After reading the manuscript, I had a few questions and comments about the work.

1.      There is a misprint in the title of the article.

2.      It is desirable to specify in the methods section how the parameter «protection of Lipid Biomarkers by minerals» was calculated.

3.      Fig.1 What do the error bars in Figures 1. mean, how many repetitions were made for each experiment?

4.      Figure 2. A typo in the scale caption of the left figure. It should be corrected to NS.

5.      lines 188-196. The authors write that the main reason for the enrichment of the silt fraction of 13C in NS soils is adsorption of microbes that selectively absorb 13C. Why such mechanism is not realised in TP soils in SS and DS?

6.      Line 240. On what basis did the authors assume that fresh SOM predominates in the SS and DS of NC soil? It is more logical to assume that fresh SOM should predominate in the topsoil.

7.      Fig. 4: Changes in CPIAF≥20 value in soil TP compared to soil NS are observed not only in sample TS, but also in samples SS and DS. How can this be explained?

8.      What is the basis for the authors' assumption that the degradation of SOM previously adsorbed on small particles may be accelerated during agricultural activities?

Reviewer 2 Report

Comments and Suggestions for Authors

The protection of soil organic matter is a very important issue, taking into account the rate of its decomposition as a result of intensive cultivation and, at the same time, a significant reduction in the use of natural fertilizers, as well as the effects of CO2 emissions into the atmosphere. For this reason, the topic of the research undertaken is current. However, as I understood from Material and Methods, the assessed paper is based on very narrow research material as for a scientific article. The description in the Material and Methods shows that the research material consists of two soil samples taken from different locations - cultivated soil and natural soil. One sample was taken from these sites from three layers: 0-20; 20-40 and 40-60 cm. Finally, a total of 6 samples were taken. If this was the case, it is insufficient material to draw conclusions and transfer these relationships to soils in general. One can only conclude that this applies to these two locations. The assessment of the significance of differences between soils and layers was based on the results of analyzes of repetitions (how many were not specified) from the same sample. This is an incorrect procedure in the analysis of experimental data, because weighments made from a correctly collected and prepared homogeneous sample are only used to assess the correctness performance of the analysis and not for statistical analysis of the results. The error bars marked in the figures and error letter markings in tables only indicate the correctness of the analysis, but not the significance of differences in the properties of the compared soils.

I marked comments in the text of manuscript.

Comments on the Quality of English Language

Minor editing of English language required.

Reviewer 3 Report

Comments and Suggestions for Authors

1. In my opinion, the title of the article does not reflect its essence and content. In fact, the lipid contents of the granulometric fractions of forest and rice soils in China were compared.

2. There is no description of the types of soils analyzed. The fact that they are located in approximately the same region does not mean they are genetically related and does not provide grounds for comparison. Moreover, the comparison is made between soils that have been located and developed for more than 50 years under completely different redox conditions (under rice under conditions of periodic long-term complete flooding and under deciduous forest)

3. The authors do not provide data on determining the mineralogical composition of fractions of different sizes. In my opinion, if we are talking about soil formation and the protective functions of minerals, first of all you need to know which minerals we are talking about.

4. General characteristics of the analyzed samples, in particular the data from Table S1 and Figure 5, as well as a discussion of the differences for the studied soils, should be given at the beginning of the article, either when describing the object, or in the results and discussion paragraph.

5. It is not clear how the isolated fractions were prepared for extraction with organic extractants and for treatment with a mixture of acids. Were they further ground to some standard particle size, or were the results obtained from treatments of different particle sizes analyzed and compared?

6. To remove the mineral part (silicate, mainly) a mixture of hydrofluoric and hydrochloric acids in high concentration was used, the treatment was carried out multiple times, the supernatant liquid, if I understand correctly, was discarded.

At the same time, there is a lot of data in the literature on the effect of such treatment on the structure of soil organic matter. Could the additional lipid content detected be an artifact of the treatments performed? Evidence needed, more research needed

7. Regarding the C13 data, since during acid treatments the supernatant solution was drained, the loss of SOM was not taken into account. Can the data presented be compared and trusted?

8. The drawings are made carelessly, the designation of the ordinate scale (Fig. 5) is covered by the caption to the scale

Round 2

Reviewer 2 Report

Comments and Suggestions for Authors

Please open file with comments to revised version of the paper.

Reviewer 3 Report

Comments and Suggestions for Authors

I believe that now, after the changes have been made, in particular, clarification of the actual purpose of the research, a clearer description of objects and methods, and changes in the structure of the presentation, the article can be published

Author Response

We greatly appreciate the time and effort you have spent processing our manuscript (agronomy-2945442) entitled “Long-Term Cultivation Results in Soil Organic Matter Degradation in Small-Particle Fractions”. Thank you very much for the comments and revision suggestions, these were vary helpful for improving the quality of our manuscript. Thank you again!